# Discovery of a small-molecule inhibitor that traps Polθ on DNA and synergizes with PARP inhibitors

William Fried[1,11], Mrityunjay Tyagi[2,11], Leonid Minakhin[2], Gurushankar Chandramouly[2], Taylor Tredinnick [2], Mercy Ramanjulu[3], William Auerbacher[2], Marissa Calbert [2,4], Timur Rusanov[5], Trung Hoang[6], Nikita Borisonnik[7], Robert Betsch[8], John J. Krais [8], Yifan Wang[8], Umeshkumar M. Vekariya[4,9], John Gordon[4], George Morton[10], Tatiana Kent[2], Tomasz Skorski[4,9], Neil Johnson [8], Wayne Childers[3,10], Xiaojiang S. Chen [1,3] & Richard T. Pomerantz [2,3] ✉

The DNA damage response (DDR) protein DNA Polymerase θ (Polθ) is synthetic lethal with homologous recombination (HR) factors and is therefore a promising drug target in BRCA1/2 mutant cancers. We discover an allosteric Polθ inhibitor (Polθi) class with 4–6 nM IC$_{50}$ that selectively kills HR-deficient cells and acts synergistically with PARP inhibitors (PARPi) in multiple genetic backgrounds. X-ray crystallography and biochemistry reveal that Polθi selectively inhibits Polθ polymerase (Polθ-pol) in the closed conformation on B-form DNA/DNA via an induced fit mechanism. In contrast, Polθi fails to inhibit Polθ-pol catalytic activity on A-form DNA/RNA in which the enzyme binds in the open configuration. Remarkably, Polθi binding to the Polθ-pol:DNA/DNA closed complex traps the polymerase on DNA for more than forty minutes which elucidates the inhibitory mechanism of action. These data reveal a unique small-molecule DNA polymerase:DNA trapping mechanism that induces synthetic lethality in HR-deficient cells and potentiates the activity of PARPi.

DNA damage response (DDR) factors facilitate DNA repair and confer resistance to DNA damaging cancer therapies and are therefore important anti-cancer drug targets[1–3]. Particular DDR factors, such as Poly ADP polymerase 1 (PARP1), are synthetic lethal with homologous recombination (HR) factors such as BRCA1 and BRCA2[2,4]. By exploiting this synthetic lethality, PARP1 inhibitors (PARPi) can selectively kill HR-deficient cancers such as those that occur in the breast, ovary, prostate and pancreas[2,4–7]. Although PARPi have revolutionized precision oncology by leveraging synthetic lethality to target HR-deficient cancers, not all patients respond to PARPi and drug resistance is a major

[1]Molecular and Computational Biology, Department of Biological Sciences and Chemistry, University of Southern California, Los Angeles, CA, USA. [2]Department of Biochemistry and Molecular Biology, Sidney Kimmel Cancer Center, Thomas Jefferson University, Philadelphia, PA 19107, USA. [3]Recombination Therapeutics, Pennsylvania Biotechnology Center, Doylestown, PA 18902, USA. [4]Fels Cancer Institute for Personalized Medicine, Philadelphia, PA, USA. [5]Department of Pharmacology and Regenerative Medicine, University of Illinois at Chicago, Chicago, IL 60612, USA. [6]Janssen Biotech, Malvern, PA 19355, USA. [7]Polysciences, Inc., Huntingdon Valley, PA, USA. [8]Nuclear Dynamics Program, Fox Chase Cancer Center, Philadelphia, PA 19111, USA. [9]Department of Cancer and Cellular Biology, Temple University Lewis Katz School of Medicine, Philadelphia, PA, USA. [10]Moulder Center for Drug Discovery Research, Temple University School of Pharmacy, Philadelphia, PA, USA. [11]These authors contributed equally: William Fried, Mrityunjay Tyagi. ✉e-mail: richard.pomerantz@jefferson.edu

problem[4,8,9]. Thus, the development of second-generation precision medicines that can simultaneously target HR-deficient cells while overcoming PARPi resistance is urgently needed for improving the survival rates of patients with HR-deficient cancers.

In 2015, it was discovered that the DDR factor DNA polymerase θ (Polθ) is synthetic lethal with BRCA1 or BRCA2 (BRCA), generating interest in this DDR protein as a promising precision oncology drug target[10,11]. Polθ is a unique DNA polymerase (Pol) and DNA helicase fusion protein that promotes multiple DNA repair mechanisms. Seminal studies in invertebrates were the first to elucidate a role for Polθ in error-prone repair of DNA double-strand breaks (DSBs) in a pathway referred to as Theta-Mediated End-Joining (TMEJ), or alternative DNA end-joining (Alt-EJ), whereby Polθ utilizes short tracts of microhomology flanking DNA breaks for facilitating DNA repair[12–14]. Subsequent biochemical studies and cellular studies in mammalian cells further elucidated Polθ DNA end-joining activity in higher eukaryotes. For example, biochemical studies discovered the ability of Polθ polymerase (Polθ-pol) to facilitate microhomology-mediated end-joining (MMEJ) of model DNA breaks with 3' single-strand DNA (ssDNA) overhangs in the absence of co-factors[15]. Studies using murine cells confirmed that Polθ error-prone DNA end-joining preferentially uses microhomology flanking DNA break sites and occurs independently of non-homologous end-joining (NHEJ)[10,16,17]. As stated above, Polθ was found to be synthetic lethal with BRCA1/2 in breast and ovarian cancer cells lines[10,11], and more recent reports confirmed this synthetic lethal relationship in additional cell lines[18,19]. Suppression of Polθ expression was also found to significantly reduced the survival of HR-deficient cells treated with PARPi, suggesting that Polθ confers resistance to PARPi[11,20]. These seminal studies revealed Polθ as a promising drug target in HR-deficient cancers and identified its central role in MMEJ. Additional studies found that Polθ confers resistance to ionizing radiation, bleomycin, cisplatin, topisomerase inhibitors and DNA crosslinking agents, indicating that Polθ inhibitors can have broad applicability in suppressing resistance to many DNA damaging cancer therapies[11,16,20–22].

The majority of Polθ's DNA repair activities require it's carboxy-terminal A-family DNA polymerase domain (Polθ-pol). For example, the DNA synthesis activity of Polθ is essential for its functions in MMEJ (TMEJ, Alt-EJ), translesion synthesis, and ssDNA gap repair[15,23–28]. Polθ-pol is closely related to widely studied A-family members *Thermus aquaticus* (Taq) Pol and *E. coli* DNA pol I Klenow fragment[29]. However, in contrast to most A-family members, Polθ-pol is highly error-prone and promiscuous which is due in part to its unique loop domains and residues that strongly interact with the primer strand[15,25,30]. The cellular function for the amino-terminal super-family 2 (SF2) DNA helicase domain of Polθ (Polθ-hel) is less well understood, but has been shown to contribute to MMEJ and the survival of BRCA1-deficient cells[31]. Biochemical studies revealed Polθ-hel ATP-dependent 3′–5′ DNA unwinding and RPA dissociation activities[31,32], and Polθ-hel has also been implicated in RAD51 binding and dissociation of RAD51-ssDNA nucleoprotein complexes[11].

Recent reports validated Polθ as a druggable target and the first Polθi has entered clinical trials. Specifically, a potent and selective allosteric Polθ-pol inhibitor class was shown to preferentially kill BRCA-deficient cancer cells and selectively induce DNA damage in these cells[33,34]. Additional data demonstrated the ability of this Polθi class to overcome PARPi resistance in BRCA1-mutant cells with a specific genetic deficiency in the SHLD/53BP1 complex that restores HR by promoting hyper 5′−3′ DNA end resection[33]. A related Polθ-pol allosteric inhibitor was recently shown to selectively kill BRCA2-null HCT116 cells and exhibit moderate in vivo efficacy as a single agent against BRCA2-null HCT116 xenografts[35]. In addition to these Polθi, the antibiotic Novobiocin was repurposed as a Polθ-hel inhibitor and demonstrated in vivo efficacy against HR-deficient xenografts as a single agent and in combination with PARPi[36]. Taken together, Polθ

continues to be an exciting and promising precision oncology drug target and Polθi are also likely to be evaluated as radiosensitizers in future clinical studies[37].

Here, we discover and characterize an allosteric Polθ-pol inhibitor class (RTx-161/RTx-152) that exhibits 4−6 nM $IC_{50}$, selectively kills HR-deficient cancer cells, and suppresses PARP inhibitor (PARPi) resistance in multiple genetic backgrounds, including HR-proficient cells. X-ray crystallography and biochemistry reveal that our Polθi class selectively inhibits Polθ-pol in the closed conformation on B-form DNA via an induced fit mechanism. In contrast, the Polθi fails to inhibit Polθ-pol catalytic activity on A-form DNA/RNA in which the enzyme binds in an open configuration. Remarkably, Polθi binding to the post-catalytic Polθ-pol:DNA/DNA closed complex traps the polymerase on DNA for >40 min which underlies the inhibitory mechanism of action. These data elucidate a unique small-molecule DNA polymerase:DNA trapping mechanism that induces synthetic lethality in HR-deficient cells and suppresses PARPi resistance in HR-deficient and HR-proficient genetic backgrounds.

## Results

### Development of a potent Polθi class

A selected hit (MC28003) from a small-molecule high-throughput screen was advanced via medicinal chemistry to yield MC360385 which improved potency by -18-fold (9 nM $IC_{50}$) against Polθ-pol (Fig. 1a). A fluorescence based Polθ-pol DNA synthesis assay was used to measure $IC_{50}$ (Fig. 1b). Here, Polθ-pol strand displacement activity results in a significant increase in Cy5 fluorescence due to dissociation of the DNA strand containing a black-hole quencher. We confirmed that MC160385 inhibits the DNA synthesis activity of recombinant human full-length Polθ (Fl-Polθ) which was characterized in our prior studies (Supplementary Fig. 1a)[23]. Considering that Polθ DNA synthesis activity is essential for its MMEJ function (Fig. 1c)[10,15], we tested MC160385 inhibition of Polθ-pol MMEJ activity in vitro using a previously characterized MMEJ assay[15,23]. As expected, MC160385 showed substantial Polθ-pol MMEJ inhibition (Fig. 1d). Biochemical assays measuring the relative velocity of Polθ-pol deoxyadenosine monophosphate (dAMP) incorporation on a traditional primer-template showed that MC160385 significantly reduces Vmax while slightly reducing Km for the 3′-deoxyadenosine triphosphate (dATP) substrate (Fig. 1e). These data suggested an uncompetitive allosteric mechanism of inhibition which was confirmed by X-ray crystallography for our Polθi class (see below). Despite the relatively high potency (9 nM $IC_{50}$) of MC160385, the compound failed to show selective killing of *BRCA2*-null DLD1 cells in colony survival assays (Supplementary Fig. 1b). This may be due to strong efflux of MC160385 by P-glycoproteins and/or extremely poor membrane permeability due to high lipophilicity (logP -5.3). Olaparib showed selective killing of *BRCA2*-null DLD1 cells as a positive control for inducing synthetic lethality (Supplementary Fig. 1c).

The addition of a polar 5-member heterocyclic ring yielded RTx-152 and RTx-161 with improved potency (-4−6 nM $IC_{50}$)(Fig. 1f, g; Supplementary Fig. 1d). As a comparison, RTx-161 showed higher potency than recently published Polθ-pol inhibitors ART558 (11.4 nM $IC_{50}$) and RP6685 (6.9 nM $IC_{50}$) using the identical assay (Supplementary Fig. 1e). RTx-161 showed selective killing of *BRCA2*-null HCT116 and DLD1 cells, with little to no effect in *BRCA2*-WT cells (Fig. 1h, i). As a comparison, ART558 and RP6685 showed slightly reduced potency in *BRCA2*-null cells (Supplementary Fig. 1f). The closely related analog of RTx-161, RTx-152 which lacks the $CH_2$-OH side-chain and has a slightly higher in vitro $IC_{50}$ (6 nM), also showed selective killing of *BRCA2*-null cells, albeit with a moderately higher $IC_{50}$ relative to RTx-161 (Supplementary Fig. 1g). The efficacy of RTx-161 was therefore further examined in multiple HR-deficient cells. RTx-161 caused a significant reduction in the survival of *PALB2*-mutant EUFA1341 cells, with little to no activity against PALB2 complemented EUFA1341 cells (Fig. 1j).

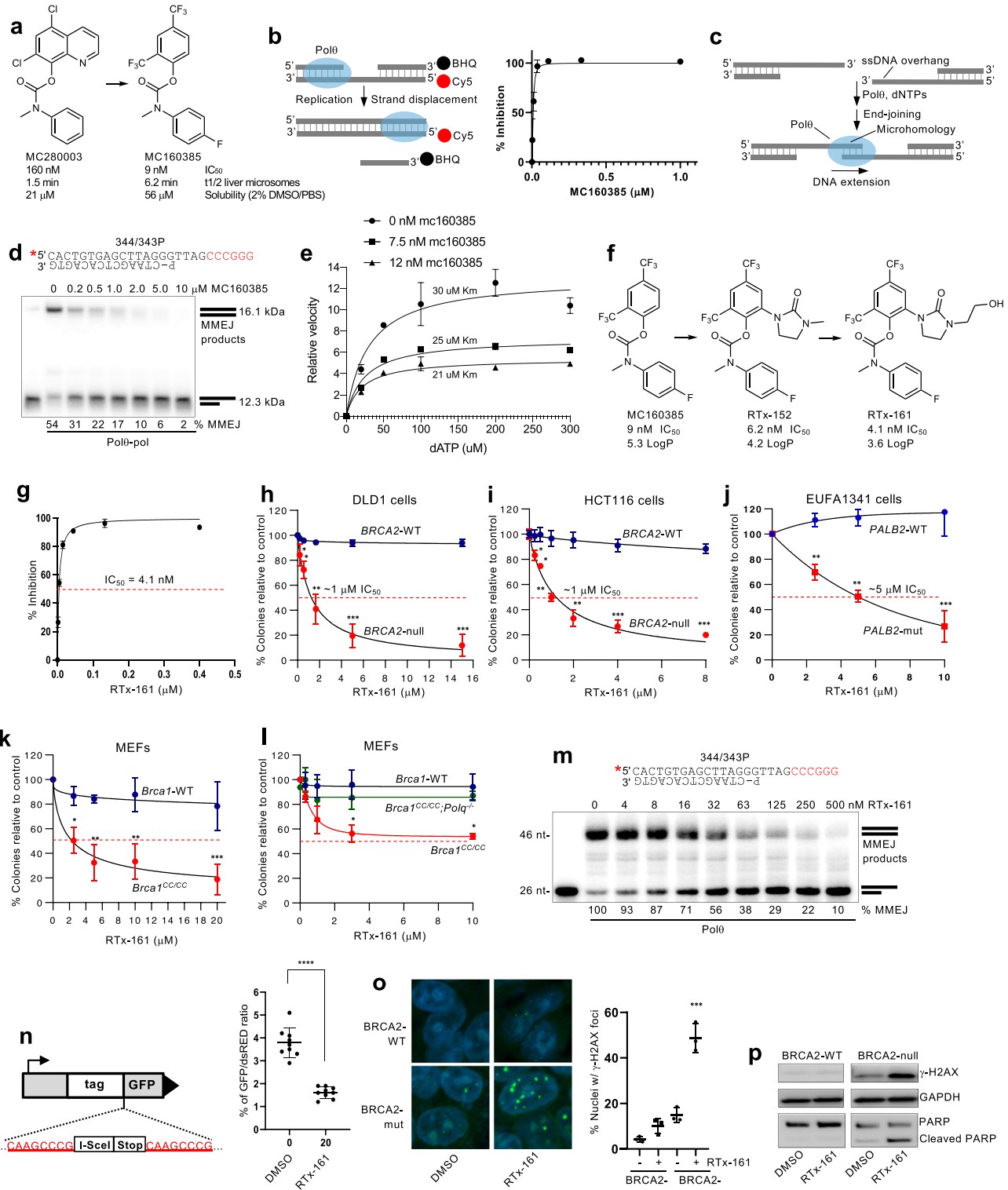

PALB2 functions with BRCA1 and BRCA2 during HR, and PALB2 mutations have recently been validated as biomarkers for HR-deficiency in breast cancer[38]. RTx-161 treatment also induced selective killing of mouse embryonic fibroblasts (MEFs) harboring a *Brca1* mutation (*Brca1cc/cc*) that is defective in PALB2 binding (Fig. 1k)[39].

To evaluate the selective on-target killing effect of RTx-161, we tested its activity against *Brca1cc/cc;Polq-/-* cells which were selected after CRISPR-Cas9 knockout of *Polq*. The results showed that *Brca1cc/cc;Polq-/-* cells were resistant to RTx-161 compared to *Brca1cc/cc* cells which indicates a selective on-target effect (Fig. 1l). Consistent with

this result, RTx-161 exhibited selective inhibition against recombinant Polθ-pol; no inhibition of six other recombinant eukaryotic Pols was observed, including the related A-family Polγ (Supplementary Fig. 1h).

Considering the relatively high potency and selectivity of RTx-161 against Polθ-pol, we further characterized its biochemical and cellular activity. As expected, RTx-161 showed concentration dependent inhibition of Polθ-pol MMEJ activity in vitro (Fig. 1m). We next used a previously characterized GFP MMEJ reporter chromosomally integrated in U2OS cells that measures MMEJ (Alt-EJ) repair of a site-specific DSB within a GFP expression cassette that is induced by

**Fig. 1 | Development and characterization of a Polθ inhibitor class. a** Structures of Polθi. **b** Strand displacement assay (left). Scatter plot showing inhibition curve of MC160385. Data represent mean. $n = 3$ (technical replicates) +/-s.d. **c** Schematic of Polθ MMEJ. **d** Schematic of MMEJ DNA (top). Non-denaturing gel showing inhibition of Polθ-pol MMEJ by MC160385 (Bottom). % MMEJ indicated. $n = 2$ (performed twice). **e** Scatter plot showing relative velocity of dAMP incorporation by Polθ-pol in the presence of the indicated concentrations of MC160385. Data represent mean. $n = 2$ (technical replicates) +/- s.d. **f** Structures of improved Polθi. **g** Scatter plot showing inhibition curve of RTx-161. Data represent mean. $n = 3$ (technical replicates) +/- .s.d. **h–l** Scatter plots showing clonogenic survival following treatment with RTx-161 relative to DMSO. Data represent mean of 3 independent experiments ($n = 3$ biological replicates) performed in triplicate ± SEM *$p < 0.05$, **$p < 0.01$, ***$p < 0.001$. Two-sample $t$-test and $P$-values are indicated. **h** $n = 3$, $P = 0.000102$ for 5 μM, DLD1 *BRCA2* -/- vs DLD1 Parental; **i** $n = 3$, $P = 0.000091$ for 4 μM HCT116 *BRCA2* -/- vs HCT116 Parental; **j** $n = 3$, $P = 0.001193$ for 5 μM EUFA1341*PALB2* -/-

vs EUFA1341 Parental; **k** $n = 3$, $P = 0.004734$ for 10 μM *Brca1$^{CC/CC}$* vs *Brca1*-WT MEFs; **l** $n = 2$, $P = 0.016209$ for 10 μM *Brca1$^{CC/CC}$* vs *Brca1*-WT MEFs, $P$ = not significant for 10 μM *Polq*-/-vs *Brca1*-WT MEFs. **m** Schematic of MMEJ DNA (top). Microhomology, red. Denaturing gel showing RTx-161 inhibition of Polθ-pol MMEJ (bottom). $n = 1$ (performed once). **n** Schematic of MMEJ reporter (left). Bar plot showing reduction in cellular MMEJ after treatment with 20 μM RTx-161 (right). Data represent mean from 3 biological replicates performed in triplicate. +/-s.d. $n = 3$, $P < 0.0001$ for 20 μM RTx-161. **o** Representative images of γ-H2AX phosphorylation following DMSO and RTx-161 treatment in DLD1 *BRCA2* -/- or DLD1 Parental cells (left). Scatter plot shows percentage of nuclei with γ-H2AX foci. $n = 3$ (biological replicates), $P = 0.000369$ for RTx-161 treated, DLD1 *BRCA2* -/- vs DLD1 Parental. **p** Western blot showing γ-H2AX phosphorylation and PARP cleavage following DMSO or RTx-161 treatment. GAPDH is shown as loading control. Source Data are provided as a Source data file.

transient expression of I-SceI endonuclease[40]. We observed RTx-161 suppression of MMEJ in U2OS cells (Fig. 1n). Our recent studies demonstrated that Polλ also promotes MMEJ repair of this GFP reporter which explains the remaining GFP cells following inhibition of Polθ[41]. Finally, we found that RTx-161 treatment selectively caused DNA damage in *BRCA2*-null DLD1 cells, indicated by phosphorylation of γ-H2AX which was detected by immunofluorescence and Western blot (Fig. 1o, p). In addition, RTx-161 selectively induced PARP cleavage and apoptosis in *BRCA2*-null DLD1 cells (Fig. 1p; Supplementary Fig. 1i). Taken together, the data presented in Fig. 1 and Supplementary Fig. 1 characterize RTx-161 as a potent and selective Polθi that exhibits preferential killing of HR-deficient cells.

## Polθi class suppresses PARPi resistance

The development of drugs that can suppress or overcome PARPi resistance in cancer cells with little to no toxicity against non-cancerous cells is expected to significantly improve the outcome of patients receiving PARPi therapies. A previously developed Polθi (ART558) targeting Polθ-pol suppressed PARPi resistance in *BRCA1*-mutant cells genetically engineered to contain specific knockouts of the Shieldin complex[33]. Inactivation of the Shieldin complex reactivates HR by promoting 5′−3′ DNA end resection at DSBs. Whether small-molecule inhibition of Polθ-pol can suppress or overcome PARPi resistance in other genetic backgrounds remains unclear.

Integrative clinical genomics of advanced prostate cancer revealed that >10% of metastatic castration-resistant prostate cancers (mCRPCs) exhibit biallelic loss of *BRCA2*[42], and Olaparib has been approved to treat BRCA-mutant mCRPC[6]. Thus, we initially examined whether RTx-161 can overcome cellular resistance to Olaparib in *BRCA2*-null cells which is a relevant model for biallelic *BRCA2* loss. Remarkably, RTx-161 exhibited strong synergistic activity with Olaparib in *BRCA2*-null HCT116 cells, and the addition of RTx-161 essentially overcome cellular resistance to Olaparib (Fig. 2a). Similar synergistic activity between RTx-161 and Olaparib was observed in *BRCA2*-null DLD1 cells (Fig. 2b). RTx-161 also exhibited strong synergistic activity with Rucaparib (Supplementary Fig. 1j). As expected, the closely related Polθi RTx-152 also exhibited synergistic activity with Olaparib in *BRCA2*-null cells (Supplementary Fig. 1k).

We next examined whether RTx-161 reduced cellular resistance to Olaparib in cell lines harboring *BRCA2* truncating mutations and pathogenic mutations. Remarkably, although RTx-161 showed little to no activity as a single agent in PE01 ovarian cancer cells harboring a single *BRCA2* mutant allele (*BRCA2.5193 C > G*), it strongly potentiated the effects of Olaparib and the combination was synergistic (Fig. 2c). RTx-161 significantly reduced cellular resistance to Olaparib in VC8 Chinese hamster cells which also possess a *BRCA2* truncating mutation (Fig. 2d). Olaparib is also approved to treat germline BRCA-mutant pancreatic cancer and, consistently, RTx-161 significantly reduced cellular resistance to Olaparib in the pancreatic cancer cell line

CAPAN-1, which harbors the *BRCA2* 6174delT pathogenic deletion mutation and loss of the wild-type *BRCA2* allele (Fig. 2e). Strong synergy was again observed between RTx-161 and Olaparib. Considering the on-target effects of RTx-161 shown in Fig. 1l and Supplementary Fig. 1f, these data reveal that Polθ-pol enzymatic activity is responsible for promoting cellular resistance to PARPi in *BRCA2*-mutant cells. Finally, we examined whether RTx-161 reduces PARPi resistance in the *BRCA*-WT and HR-proficient triple negative breast cancer cell line MDA-MB-231. Remarkably, RTx-161 modestly potentiated the effects of Talazoparib in MDA-MB-231 cells despite their HR-proficient status (Fig. 2f). These data demonstrate that Polθ-pol enzymatic activity also contributes to PARPi resistance in HR-proficient cells, albeit to a lesser degree than in HR-deficient cells. Since reactivation of HR in BRCA-deficient cells is a common mechanism of PARPi resistance, these data indicate another advantage for combining PARPi with Polθi[8]. Taken together, these results demonstrate that RTx-161 inhibition of Polθ-pol strongly sensitizes HR-deficient cells to PARPi, and modestly potentiates the effects of PARPi in HR-proficient cells.

## Structural basis for Polθ small-molecule inhibition

To elucidate the mechanism of action of our Polθi class, we investigated its structural basis of inhibition using X-ray crystallography. A previously optimized highly soluble Polθ-pol construct (PolθΔL) was used for X-ray crystallography in which five short disordered loops were replaced with serine-glycine linkers[43]. We confirmed that the PolθΔL construct is inhibited by our Polθi class (Supplementary Fig. 2a). Co-crystals of a quaternary complex of PolθΔL bound to a DNA/DNA primer-template, RTx-161 and incoming ddGTP did not diffract well. As an alternative approach, we solved the quaternary structure of PolθΔL bound to the closely related analog RTx-152 (Fig. 3a). We conducted co-crystallization experiments using PolθΔL, along with RTX-152, DNA/DNA primer-template with a 5′ 10 nt ssDNA overhang, and incoming 2′,3′-dideoxyguanosine triphosphate (ddGTP). The structure of the resulting complex was determined to a resolution of 3.24 Å (Table 1, Fig. 3a). The co-crystal structure of PolθΔL complexed with DNA/DNA, ddGTP, and RTx-152 shows the enzyme in the closed conformation with the inhibitor residing in an allosteric binding pocket located in the fingers domain of the polymerase (Fig. 3a, Supplementary Fig. 2b). The location of the inhibitor binding pocket is similar to those observed in previously published Polθ-pol inhibitor structures[34,35]. Within the binding pocket, RTx-152 forms hydrophobic contacts with many of the surrounding residues and a hydrogen bond with Y2420 (Fig. 3b, Supplementary Fig. 2c–e). The electrostatic surface charge representation shows the inhibitor binding pocket has either a neutral or slightly positive electrostatic potential surface (Fig. 3c).

Comparison of the allosteric binding pocket of the PolθΔL complex (green) with the previously solved structure of Polθ-pol in the closed

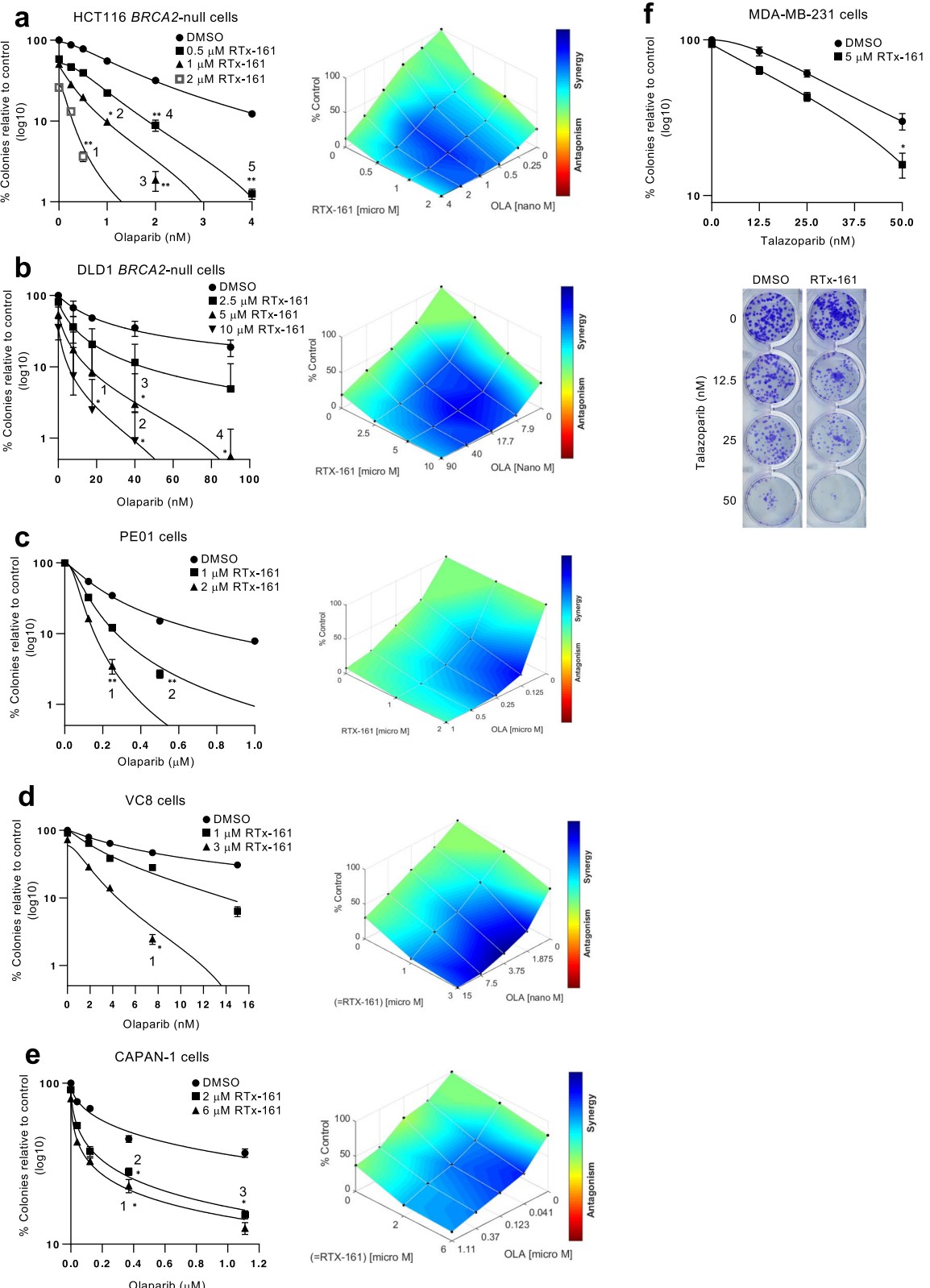

**Fig. 2 | Polθi acts synergistically with PARPi. a–e** Scatter plots showing clonogenic survival of the indicated cell lines following treatment with the indicated concentrations Olaparib and RTx-161 relative to DMSO controls (left). Percentage of colonies are normalized to DMSO treated cells (DMSO = 100%). Data represent mean."*n*" denotes number of independent experiments with triplicates for each condition, ±SEM *p < 0.05, **p < 0.01, ***p < 0.001. Statistical significance was measured from two-sample *t*-test and *P*-values are indicated. Synergy plots generated by Combenefit (right). **a** *n* = 3, *P*-values for 1 = 0.000793, 2 = 0.010008,

3 = 0.000135, 4 = 0.004802, 5 = 0.00832; **b** *n* = 3, *P*-values for 1 = 0.01335, 2 = 0.031029, 3 = 0.042988, 4 = 0.036713; **c** *n* = 3, *P*-values for 1 = 0.00011, 2 ≤ 0.0001; **d** *n* = 3, *P*-values for 1 = 0.021298; **e** *n* = 2, *P*-values for 1 = 0.05, 2 = 0.042428, 3 = 0.008715. **f** Scatter plot showing clonogenic survival of MDA-MB-231 cells following treatment with the indicated concentrations Talazoparib and RTx-161 relative to DMSO controls (top). Data represent mean from 3 biological replicates performed in triplicate. ±SEM. Representative images of colony plates (bottom). *P* = 0.019935 for 50 nM. Source Data are provided as a Source data file.

conformation without an inhibitor (PDB 4x0q, magenta) highlights the induced fit mechanism of RTx-152 binding (Fig. 3d). The root-mean-square deviation (RMSD) between the binding pockets of both structures (Cα atoms in residues 2330–2425) is 1.4 Å, indicating significant movement of the binding site around the inhibitor. This movement is more pronounced when focusing on the residues in and around the N alpha helix (Cα atoms in residues 2354–2376), which have an RMSD of 2.3 Å between the binding pockets before and after RTx-152 binding. The movement observed in the allosteric binding pocket upon RTx-152 binding is largely due to the engagement of several key residues located on the surrounding alpha helices that form the binding pocket (Fig. 3e–g). Upon RTx-152 binding, Y2412 shifts its position to form pi stacking with the inhibitor, which in turn engages residues W2366 and M2402 to facilitate hydrophobic packing within the binding site (Fig. 3e). The pull of the N alpha helix towards the inhibitor brings E2365 closer to R2419, allowing for the formation of a salt bridge with a distance of 3.6 Å between the two residues (Fig. 3f). The salt bridge forms over the inhibitor and blocks its exit from the binding pocket. Additionally, the bound inhibitor engages F2416 in pi stacking while pulling Y2420 in range to form a hydrogen bond (Fig. 3g).

Intriguingly, the binding of RTx-152 to the closed conformation of PolθΔL appears to lock the enzyme into this state on the DNA/DNA, which likely prevents or slows the transition back to the open conformation which is required for subsequent nucleotide binding and addition cycles. This is due to the fact that the inhibitor occupies the same space as the N and O alpha helices during the open to closed conformation shift. When comparing our PolθΔL complex structure (green) with our previously published open conformation structure of PolθΔL bound to an A-form DNA/RNA substrate (PDB: 6XBU, red), the RTx-152 binding pocked composed of the M, N, O, O$_1$, and O$_2$ alpha helices is only formed in the closed conformation (Fig. 3h). In the open conformation, the N and O helices are positioned where the bound inhibitor would be in the closed conformation (Fig. 3h). The transition of Polθ-pol from the closed to open conformation results in the O helix moving upward into the binding pocket, causing the N helix to shift by 45°, which pushes against the M helix and displaces it by ~6 Å (Figs. 3h and 4a)(Supplementary Movies 1, 2). This transition between the two conformations significantly reshapes the binding pocket, resulting in an RMSD of 8.4 Å between the open and closed states (Cα atoms in residues 2330–2425), with most of the movement occurring in the M, N, and O helices, which has an RMSD of 11.6 Å (Cα atoms in residues 2346–2391). Overall, the structural data reveal that the Polθi exclusively binds the closed conformation of Polθ-pol when the enzyme is actively engaged on DNA and suggests the inhibitor blocks the closed to open conformational change by stabilizing the enzyme: DNA/DNA complex in the closed state.

## Polθi class selectively inhibits Polθ-pol in the closed state

In prior studies, we discovered that Polθ-pol exhibits reverse transcriptase (RT) activity on DNA/RNA primer-templates and that the enzyme unexpectedly bound the wider A-form DNA/RNA in the presence of the incoming ddGTP in the open configuration (Fig. 4a, right)[43]. Despite the enzyme binding DNA/RNA in the open state, the PolθΔL:DNA/RNA:ddGTP open complex structure was solved following a single round of ddGMP incorporation and thus was catalytically active[43]. This suggests that Polθ-pol primarily performs RT activity in the open configuration. Because the hydrophobic inhibitor binding site is exclusively formed in the closed conformation (Figs. 3h, 4a, left), this strongly suggested that our compound class would be unable to inhibit Polθ-pol catalytic activity for the RT function on the DNA/RNA substrate in the open configuration. If this were the case, the Polθ:DNA/RNA open complex would be expected to be resistant to our Polθi. We examined whether RTx-161 suppressed the enzyme's relative rate of single nucleotide (ddGMP) addition on DNA/RNA (open conformation) versus DNA/DNA (closed conformation). As predicted,

RTx-161 failed to significantly inhibit Polθ-pol ddGMP incorporation on DNA/RNA in which the enzyme binds in the open configuration (Fig. 4c). In contrast, RTx-161 showed significant inhibition of Polθ-pol ddGMP incorporation on DNA/DNA in which the enzyme catalyzes phosphodiester bond formation in the closed configuration (Fig. 4b). Notably, the relative rate of Polθ-pol ddGMP incorporation was significantly slower on DNA/RNA which is consistent with defective positioning of the incoming nucleotide for the nucleotidyltransferase reaction in the open state. Control reactions showed that the rate of Polθ-pol primer extension on DNA/RNA was also significantly slower in the presence of all four 2'-deoxyribonucleotides (dNTPs)(Supplementary Fig. 3a), and the open Polθ:DNA/RNA complex was again resistant to RTx-161 under these run-off conditions (Supplementary Fig. 3b). Prior studies demonstrated that Polθ-pol exhibits a nearly identical Kd for DNA/DNA and DNA/RNA which rules out defective catalytic activity due to inefficient DNA/RNA primer-template binding[43]. Taken together, the biochemical and structural data demonstrate that our Polθi class exclusively inhibits closed Polθ-pol:DNA/DNA complexes due to specific formation of the inhibitor binding pocket within the closed configuration.

## Polθi class acts via a DNA trapping mechanism

The open-to-closed conformational change exhibited by A-family Pols during the nucleotide addition cycle has been highly characterized by structural, biochemical and biophysical studies[44–47]. For example, the polymerase binds the incoming dNTP in the open state on DNA/DNA (Fig. 5a, left). The fingers subdomain then closes ~42° upon the correctly base-paired dNTP substrate in the active site, resulting in the closed state which facilitates phosphodiester bond formation (dNMP addition)(Fig. 5a, center). In order to bind the next incoming nucleotide and continue the replication cycle, the enzyme must transition back to the open state which is associated with pyrophosphate release and forward translocation (Fig. 5a, right). Because the Polθi exclusively inhibits the closed Polθ-pol:DNA/DNA complex via an allosteric mechanism, we hypothesized that our compound class blocks the closed-to-open conformational change which is supported by the X-ray structure. For instance, once the inhibitor is firmly bound within the binding pocket in the closed configuration, it appears to block the O and N helices from moving into their respective open states, which can conceivably prevent the closed-to-open transition (Figs. 3h and 4a). We envisaged that this allosteric inhibitory mechanism would additionally trap the enzyme in the closed state on DNA and thus prevent dissociation of the closed Polθ-pol:DNA/DNA complex.

We investigated whether our Polθi class suppresses dissociation of Polθ-pol:DNA/DNA closed complexes using endonuclease footprinting assays. Polθ-pol was incubated with a $^{32}$P-labeled primer-template containing a EcoRI recognition site within the double-strand DNA portion in the presence or absence of Polθi (Fig. 5b). Next, ddCTP or no substrate was added. Last, EcoRI was added along with an excess amount of unlabeled primer-template lacking an EcoRI site. In this scenario, once the enzyme dissociates from the initial $^{32}$P-labeled primer-template containing the EcoRI site, it will primarily bind the excess unlabeled DNA substrate as a trap, allowing for unhindered EcoRI endonuclease activity on the initial substrate. Reactions were terminated at various time intervals by the addition of EDTA, and radiolabeled DNA was analyzed via denaturing polyacrylamide gel electrophoresis (PAGE) and phosphorimager analysis. As a control, we demonstrate a time course of EcoRI cleavage in the absence of Polθ-pol and inhibitor (Fig. 5c, lanes 1–4; middle plot). Nearly identical rates of EcoRI cleavage were observed when Polθ-pol was pre-incubated with the primer-template with or without ddCTP (Fig. 5c, lanes 5–10; middle plot). These data demonstrate that Polθ-pol readily dissociates from the $^{32}$P-labeled primer-template with or without ddCTP present, which enables efficient EcoRI cleavage. The results also show that Polθ-pol extended the primer stand by one nucleotide due to the addition of

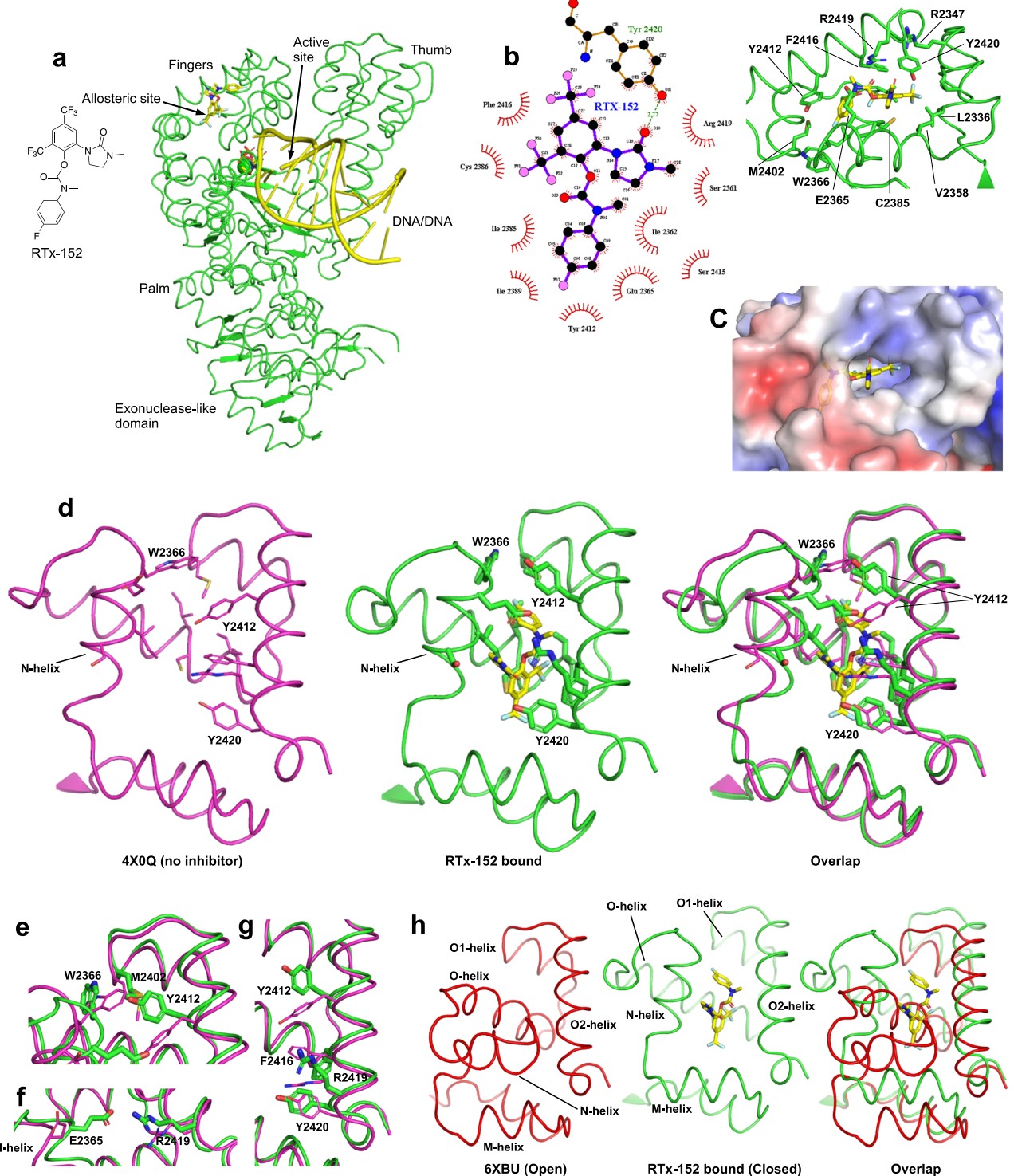

**Fig. 3 | Structural basis of Polθ-pol inhibition. a** Overall structure of PolθΔL:DNA/DNA:ddGTP:RTx-152. **b** Left side of figure is a 2D ligand plot map detailing the interactions between inhibitor RTx-152 and the surrounding allosteric pocket. Hydrophobic contacts are shown by the red radiating symbols while the hydrogen bond between Y2420 and O20 of RTx-152 is shown by a green dashed line. The right side of the figure is a 3D model in PyMOL detailing the position of each of the residues shown on the left within the allosteric binding site in relation to RTx-152. **c** Electrostatic surface representation of PolθΔL at the allosteric inhibitor binding site. Positive potential is in blue, negative potential is in red, and neutral in white. **d** 3D structure of the allosteric binding pocket of both the previously published closed conformations of Polθ-pol (PDB 4x0q, in magenta) and that of the PolθΔL RTx-152 inhibitor complex in this study (in green). Shown here and in (**e**–**g**) are differences in key residues due to the induced fit mechanism of the binding of RTx-152. **e** RTx-152 binding to the pocket induced Y2412 switch to form pi stacking, which in turn to induce W2366 switch to form a hydrophobic packing. **f** RTx-152 binding induced the conformational switch of E2365 and R2419 to form a salt bridge across surface of the binding pocket. **g** The bound RTx-152 engages key residues such as Y2412, F2416 and Y2420 on alpha helix O2, forming pi stacking with Y2412 and F2416 and a hydrogen bond with Y2420. **h** Overlap of the allosteric RTx-152-binding pockets between the open and closed conformations of PolθΔL, showing that the pocket in the closed state is not present in the open state.

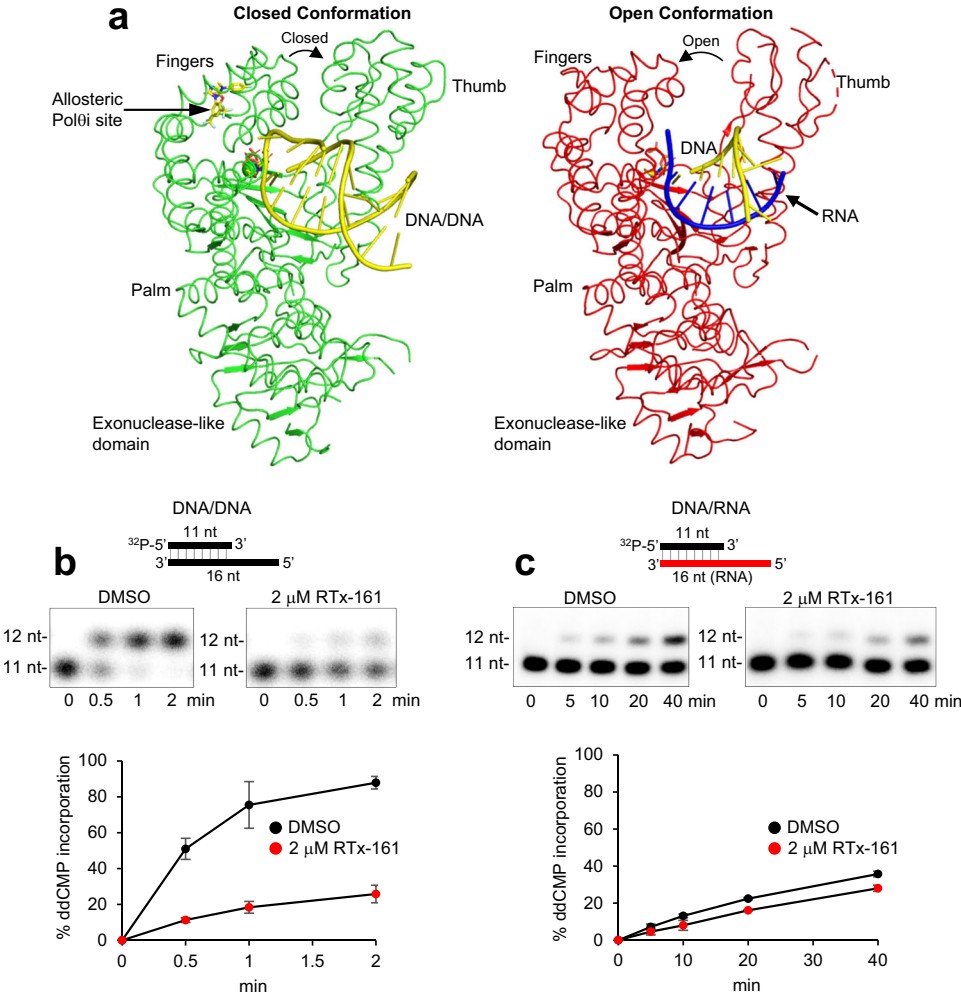

**Fig. 4 | Polθi exclusively inhibits closed Polθ-pol:DNA/DNA complexes. a** Side-by-side comparison of the open and closed conformations of PolθΔL. DNA and RNA are indicated as yellow and blue, respectively. **b, c** Schematic of DNA/DNA (**b**) and DNA/RNA (**c**) templates (top). Denaturing gels showing Polθ-pol ddCMP incorporation in the presence of DMSO or RTx-161 at the given time points either on DNA/DNA (**b**) or DNA/RNA (**c**) (middle panels). Scatter plots showing relative rates of Polθ-pol ddCMP incorporation on DNA/DNA (**b**) or DNA/RNA (**c**) in the presence of DMSO or RTx-161 (bottom). Data represent mean. $n = 3$ (technical replicates) +/- s.d. Source Data are provided as a Source data file.

ddCMP as expected (Fig. 5c, lanes 8–10). The nucleotide extension product is noted as i + 1 and represents a post-catalytic complex (Fig. 5c, upper band, lanes 8–10). The unextended DNA represents a pre-catalytic complex and is noted as i (Fig. 5c, lanes 1–7, upper band). The assay was next repeated, however, RTx-161 was added along with Polθ-pol and the DNA/DNA substrate. Remarkably, the addition of RTx-161 almost completely suppressed EcoRI DNA cleavage for >10 min (Fig. 5c, lanes 11–13, middle plot). As controls, we show that RTx-161 and RTx-152 do not inhibit EcoRI activity in the absence of Polθ-pol (Supplementary Fig. 4a). Hence, binding of RTx-161 to the Polθ-pol:DNA/DNA closed complex prevents dissociation of the polymerase from the primer-template which in turn physically blocks EcoRI endonuclease activity. Intriguingly, we observed that the small fraction of cleaved DNA/DNA was derived primarily from the pre-catalytic Polθ-pol:DNA/DNA complex (i; Fig. 5c, lanes 11–13). For example, quantitation of the disappearance of the DNA species represented by the post-catalytic (i + 1) and pre-catalytic (i) complexes showed a significant decrease in the i species over time, with a minor increase in the i + 1 species (Fig. 5c, right plot). These data therefore demonstrate that RTx-161 exhibits significantly stronger trapping activity towards the post-catalytic complex (i + 1) since this complex is resistant to EcoRI cleavage as compared to the pre-catalytic complex (i). We observed nearly identical results with the closely related Polθi RTx-152. Here

again, the addition of RTx-152 resulted in strong suppression of endonuclease cleavage, and the post-catalytic complex (i + 1) was fully resistant to EcoRI activity (Fig. 5d). Remarkably, repeating the assay with a longer time course showed that RTx-152 traps Polθ-pol on DNA/DNA for >40 min (Supplementary Fig. 4b). The DNA species within the post-catalytic complex (i + 1) was again fully resistant to cleavage, whereas the DNA associated with the pre-catalytic complex (i) was significantly more susceptible to cleavage (Supplementary Fig. 4b). A similar trapping pattern was observed for full-length Polθ (Fl-Polθ). Here, Fl-Polθ was more strongly inhibited by RTx-152, resulting in a higher proportion of pre-catalytic complexes (i)(Fig. 5e, lanes 11–13). Despite the stronger inhibition of Fl-Polθ, the post-catalytic complexes (i + 1) were again fully resistant to EcoRI cleavage as a result of RTx-152 addition (Fig. 5e, lanes 11–13; right plot). Similar results were observed using a previously engineered active Polθ polymerase-helicase fusion construct lacking the central domain (PolθΔcen; Supplementary Fig. 4c). Additional controls show that RTx-161 and RTx-152 exhibit identical trapping of post-catalytic Polθ-pol:DNA/DNA complexes in a different sequence context (Supplementary Fig. 4d–f). These biochemical data demonstrate that our compound class suppresses dissociation of post-catalytic Polθ:DNA/DNA complexes and therefore traps the polymerase on DNA/DNA in the closed state. The observed closed state trapping mechanism supports a model in which the inhibitor blocks the closed-to-open

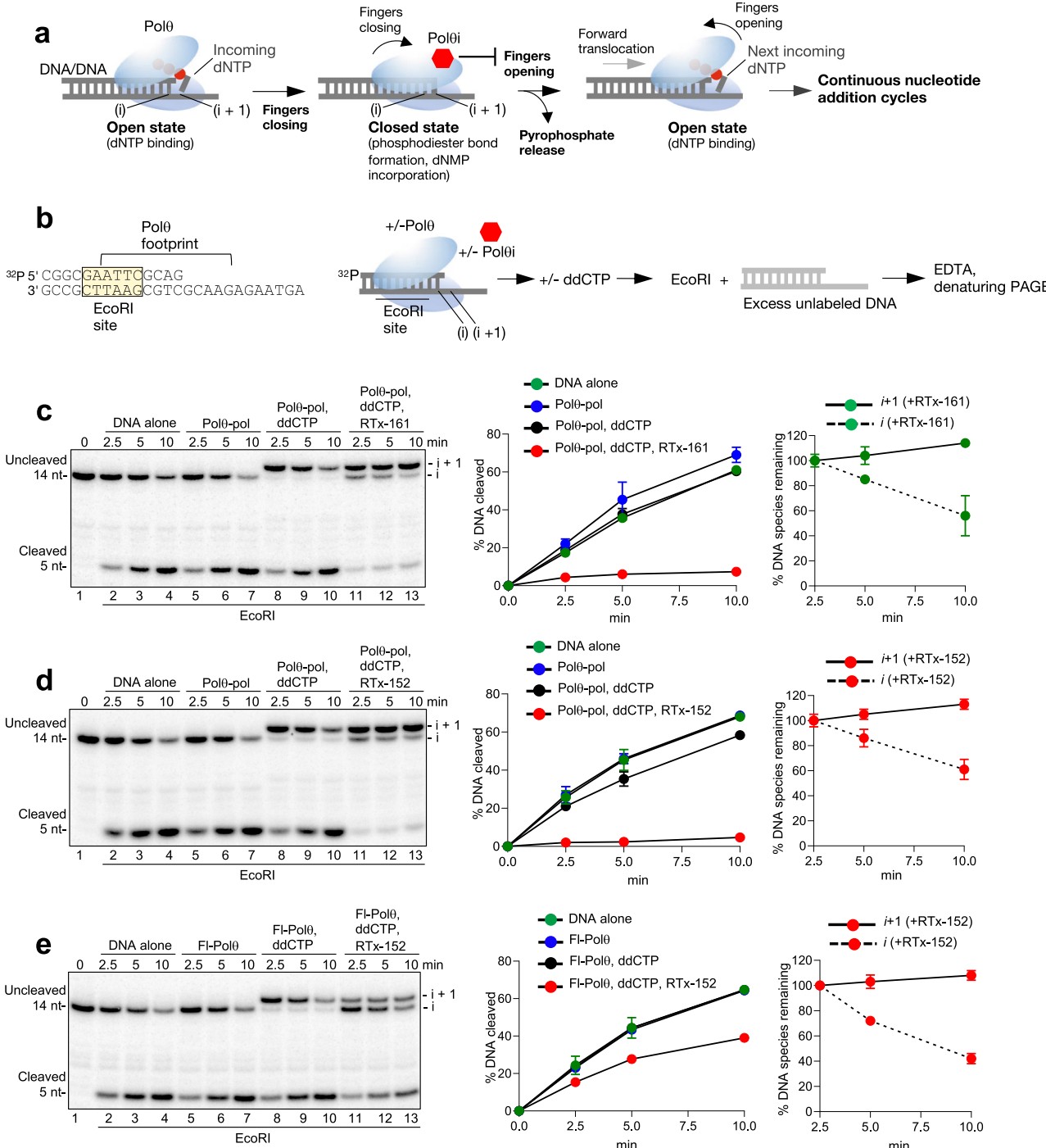

**Fig. 5 | Polθi traps Polθ-pol on DNA/DNA in the closed configuration. a** Cartoon of the open-to-closed nucleotide binding and addition cycle. The model predicts that the Polθi traps Polθ on the DNA/DNA primer-template in the closed configuration which prevents the closed-to-open transition. **b** Schematic of the DNA/DNA primer-template used for the endonuclease protection assay (left). Schematic of the EcoRI endonuclease protection assay (right). **c–e** Denaturing gel showing EcoRI cleavage of the DNA/DNA primer-template at the indicated times following pre-incubation with or without Polθ-pol (**c, d**) or Fl-Polθ (**e**) ddCTP and Polθi as

indicated (left). Scatter plot showing the relative rates of EcoRI cleavage following pre-incubation with or without Polθ-pol, ddCTP and Polθi as indicated (middle). Data represent mean. $n = 3$ +/- s.d. Scatter plot showing the relative rates of disappearance of the DNA/DNA species within the i and i + 1 complexes (right). Data represent mean. $n = 3$ (technical replicates) +/- s.d. **i** DNA within pre-catalytic complexes. i + 1, DNA within post-catalytic complexes. Source Data are provided as a Source data file.

conformational change which is essential for binding the next incoming nucleotide and continuous DNA replication (Fig. 5a).

## Discussion
PARP inhibitors have revolutionized precision oncology by leveraging synthetic lethality to target cancers with mutations in the HR pathway.

However, a significant fraction of patients fail to respond to PARPi and drug resistant is a major problem[4,8,9]. Thus, the development of second-generation precision medicines that suppress or overcome PARPi resistance while simultaneously targeting HR-deficient cancers is urgently needed. Suppression or knockout of Polθ has been shown to induce synthetic lethality in HR-deficient cells and sensitize HR-

deficient cells to PARPi[10,11,18–20]. Considering that Polθ is dispensable for HR-proficient cells and mice[10,11], it is an ideal second-generation precision oncology drug target for HR-deficient cancers.

Although prior studies have reported potent small-molecule inhibitors of Polθ-pol, and the first Polθ-pol inhibitor has entered clinical trials, the detailed mechanism by which Polθ-pol is inhibited has yet to be elucidated. Furthermore, whether Polθ-pol inhibitors can suppress or overcome PARPi resistance in various genetic backgrounds has not been explored. In this report, we discover and fully characterize a Polθ-pol inhibitor class that exhibits 4–6 nM IC$_{50}$ and induces synthetic lethality in HR-deficient cells. Our Polθi class exhibits synergistic activity with PARPi, and this effect strongly sensitizes *BRCA2*-mutant cell lines to PARPi. Additionally, our Polθi demonstrated the ability to sensitize HR-proficient breast cancer cells to PARPi. These results suggest a broad applicability of using Polθi to reduce cellular resistance to PARPi, regardless of their HR status.

The combination of biochemical and structural studies has elucidated the underlying mechanism of action of our inhibitor class. The proposed model for Polθ inhibition by our compound class involves trapping of Polθ-pol in the closed conformation after adding a nucleotide to the primer—referred to as the post-catalytic closed complex—which freezes the movement of the N and O helices of the fingers subdomain to block transition to the open conformation in the next cycle (Fig. 5a, Supplementary Movies 1, 2). For example, during the DNA repair synthesis cycle, Polθ-pol shifts from its open conformation to its closed conformation to facilitate formation of the phosphodiester bond that enables step-wise elongation of the DNA primer. As Polθ-pol transitions to its closed conformation, the allosteric binding pocket for RTx-152/RTx-161 is formed, allowing the inhibitor to enter the pocket to further change the pocket side-chain conformations to achieve an induced fit-biding (Fig. 3h, middle). Once the small-molecule enters the allosteric pocket, hydrophobic interactions between the inhibitor and the surrounding residues within the pocket cause the pocket to enclose around the inhibitor (Fig. 3d). The enclosure of the binding pocket around the inhibitor include (but not limited to) bringing E2365 and R2419 in range to form a salt bridge across the binding pocket, locking the inhibitor in place (Fig. 3f, Supplementary Fig. 2c). Additional pi stacking interactions between the inhibitor with Y2412 and F2416, as well as the formation of a hydrogen bond with Y2420, further stabilize the inhibitor within the binding pocket (Supplementary Fig. 2d). Once the inhibitor is firmly bound within the binding pocket, it prevents the conformational change of Polθ-pol back from its closed conformation to its open conformation by blocking the movement of the O and N helices of the fingers subdomain back into the open conformation. Because Polθ-pol becomes trapped in the closed conformation, it is no longer able to proceed with subsequent nucleotide addition cycles necessary for DNA repair.

Utilization of endonuclease protection assays provided further evidence that the inhibitor locks Polθ-pol in the closed state on DNA. For instance, EcoRI and KpnI protection assays revealed that post-catalytic Polθ-pol:DNA/DNA closed complexes were resistant to endonuclease cleavage exclusively in the presence of RTx-161 or RTx-152. We further found that such closed Polθ-pol:DNA/DNA complexes were resistant to endonuclease cleavage for >40 min, revealing that our inhibitor class traps the polymerase on DNA/DNA in the closed complex for a prolonged period. Evidence for our inhibitor acting exclusively against Polθ-pol closed complexes was derived from biochemical data showing that catalytically active Polθ-pol:DNA/RNA open complexes were highly resistant to inhibition by our compound class, which is supported by the structural data showing that the inhibitor's binding pocket is exclusively formed in the closed configuration.

Intriguingly, small-molecule trapping of DNA repair enzymes on their respective DNA substrates appears to be a relatively common mechanism of action. Considering that DNA repair enzymes exhibit major conformational changes on DNA necessary for their respective catalytic activities, it is not surprising that effective small-molecule inhibitors act by locking DNA repair enzymes in a particular conformation while bound to DNA. For example, PARPi are well known to trap PARP:DNA complexes which is considered the major mechanism by which PARPi induce replication arrest and synthetic lethality in HR-deficient cancers[4]. Topoisomerase inhibitors are well known for their ability to covalently link their respective enzyme targets to DNA[48]. DNA methyltransferases are also known to be trapped on DNA by their nucleotide analog inhibitors[49]. In this report, we present the first evidence of small-molecule inhibitor trapping of Polθ-pol on DNA as the underlying mechanism of action of an effective Polθi class.

## Methods
### Protein purification for X-ray crystallization
The gene encoding Polθ-pol (residues 1819–2590) was cloned into a pSUMOstar vector to generate a sumo fusion that carries an N-terminal 6xHis tag separated by a PreScission protease cleavage site. This vector was then used to generate the PolθΔL construct through the in-fusion kit from Takara. The DNA fragment insert, ordered from Thermo Fisher, consisted of the codon optimized Polθ-pol sequence for bacterial protein expression. The five flexible loop regions were removed and replaced with different glycine-serine spacers dependent on the distance between the two ends of the deleted loop, which is as described previously[43]. The replacements are as follows: residues 1861–1895 were replaced by the spacer GSG, residues 1918–1934 were placed by the spacers GGSGG, residues 2146–2175 were replaced by residues GGSGG, residues 2261–2306 were replaced by the residues GGSG, and residues 2513–2526 were replaced by the residues GGSGG. The result of the In-Fusion cloning was a PolθΔL sumo fusion protein carrying an N-terminal 6xHis tag separated by a PreScission protease cleavage site.

PolθΔL-expressing *E. coli* BL21(DE3) cells were cultured at 37 °C in LB medium until optical density at 600 nm (OD$_{600}$) reached 0.3–0.4. The growth temperature was then lowered to 16 °C and *E. coli* cells were further cultured until an OD$_{600}$ of 0.7–0.9 was reached. Protein expression was then induced by the addition of 0.1 mM isopropyl β-D-thiogalactopyranoside. *E. coli* cells were cultured at 16 °C overnight and harvested by centrifugation. The cell pellet was resuspended in buffer L (50 mM Hepes pH 8.0, 500 mM NaCl, 0.005% (v/v) Igepal CA 630, and 0.5 mM TCEP); lysed by sonication in the presence of DNase I (100 μg/mL), RNase A (100 μg/mL), 10 mM MgCl$_2$, 2 mM CaCl$_2$, and 1 mM phenylmethylsulfonyl fluoride; and then centrifuged at 21612 g for 45 min. The 6 × His sumo fusion protein was captured by Ni-NTA agarose gravity-flow chromatography and then was followed by a wash consisting of 5x resin volume of buffer W (50 mM Hepes (pH 8.0), 500 mM NaCl, 0.005% (v/v) Igepal CA 630, 0.5 mM TCEP, and 10 mM imidazole). One resin volume of buffer L with 1.25% (v/v) of Precission Protease was added to the column to cleave PolθΔL from the 6×His tag through an overnight incubation at 4 °C. The cleaved PolθΔL was eluted another 2x resin volume of buffer L. The eluted protein was concentrated to about 1 mL then 0.2% Benzonase and 5 mM MgCl$_2$ was added to the protein and the mixture was incubated overnight to remove any bound DNA. PolθΔL was purified to further using a HiTrap Heparin column followed by a S200 size exclusion column (GE Healthcare Life Sciences). The protein was then concentrated to 23.9 mg/mL in a buffer of ammonium acetate (150 mM), KCl (150 mM), tris-HCl buffer (pH 8.0) (40 mM), TCEP (2.5 mM), and glycerol (1% v/v). The concentrated protein was then aliquoted and stored at −80 °C.

### Co-Crystallization
The crystallization condition for PolθΔL-pol in complex with primer/template dsDNA, ddGTP, and RTX-152 was identified by a wide matrix

screening using sitting drop plates followed by successive rounds of optimization through further grid screening with hanging drop plates. PolθΔL at a concentration of 2.5 mg/mL was mixed with a dsDNA overhang (50 μM, DNA primer: 5′-CGACGTCGCAGCGC-3′, DNA template: 5′-GCGAGACTCCGCGCTGCGACGTCG-3′, ordered and HPLC purified from IDT) and 1 mM RTX-152 in 100% DMSO, while in the presence of ddGTP (1 mM), sucrose monolaurate (300 μM), MgCl$_2$ (1 mM), and spermine tetrahydrochloride (20 mM). The final crystals used for data collection were obtained through optimization in a hanging drop tray containing a 900 μL mother liquor of 0.24 M Sodium citrate, 18% (w/v) PEG 3350, and 0.1 M Bis Tris Propane pH 7.5. Cryoprotection was achieved by looping the crystals into a mother solution containing an additional 20% glycerol before flash cooling in liquid nitrogen.

## Data collection and structural determination

Diffraction data for PolθΔL-pol complex was collected at beamline 23-ID-D at the Structural Biology Facility at the Advanced Photon Source, Argonne National Laboratory, Chicago. A complete dataset was collected using JBluIce-EPICS and processed using HKL2000 where the data was indexed, integrated, and scaled[50]. Molecular replacement was performed using the Phaser-MR program from the PHENIX package, while model rebuilding and refinement were carried out using COOT and Phenix for simulated annealing and refinement, respectively. Ligands were generated using the eLBOW program in Phenix based on their SMILES code with any incorrect bond length and angels corrected using REEL[51,52]. The initial search model for molecular replacement was the Polθ-pol structure (4x0q) with the DNA and ddGTP removed because of the difference of the DNA substrate. The initial phases of the MR protein model were improved by cyclic model building and refinement until a good model for the complex was achieved. The final model for PolθΔL-pol complex was solved at a resolution of 3.24 Å (Table 1).

## Cell lines

U2OS cells with MMEJ reporter (EJ2-GFP) was a kind gift from Dr. Jeremy Stark (City of Hope) and were generated in prior studies as described[53]. The cells were cultured in Dulbecco's Modified Eagle

Medium (DMEM, GIBCO) supplemented with 15% fetal bovine serum (Cytivia), 2 mM L-glutamine (Sigma) and penicillin/streptomycin (Sigma). DLD1 *BRCA2* -/- and DLD1 Parental were obtained from Horizon discovery, Waterbeach, UK. HCT 116 *BRCA2* -/- and HCT 116 Parental were obtained from Cancertools, London, UK. MEF *BRCA1* -/- and MEF Parental was a kind gift from Dr. Neil Johnson (Fox chase Cancer Center). EUFA1341 *PALB2* mut, *PALB2* wildtype was a kind gift from Dr. Bing Xia (Rutgers University). MDA 436 *BRCA1* mut and MDA 231 (used as wildtype control for MDA 436) cells were obtained from ATCC, Manassas, VA. DLD1 *BRCA2* -/-, DLD1 Parental, MDA 436 *BRCA1* mut and MDA 231 were cultured in RPMI supplemented with 10% fetal bovine serum, 2 mM L-glutamine, non-essential amino acids, and penicillin/streptomycin. HCT 116 *BRCA2* -/-, HCT 116 Parental, MEF *BRCA1* -/-, MEF Parental, EUFA1341 *PALB2* mut, *PALB2* wildtype were cultured in DMEM supplemented with 10% fetal bovine serum, 2 mM L-glutamine, non-essential amino acids, and penicillin/streptomycin.

## Chromosomal MMEJ GFP assay

U2OS cells harboring the chromosomal (EJ2-GFP cassette) MMEJ reporter were plated and treated with various concentrations of RTX-161 for 16 h. Cells were co-transfected with either pCMV−3x- NLS-I-SceI or control vector pCMV-3x-NLS using Lipofectamine 2000 (Invitrogen) along with dsRED-Mito cDNA (control for transfection efficiency). GFP$^+$ frequencies were measured 3 days post transfection by FACS using Facscanto (Becton Dickinson) in triplicates and corrected for transfection efficiency and background events.

## Clonogenic assay

Cells were plated on six or twelve or twenty-four well plates. In six well plate, 1000 cells/ well of *BRCA2* -/-, *BRCA1* -/-, *PALB2* mut and 500 cells/ well of wildtype pair was plated; In twelve well plate, 600 cells/ well of *BRCA2* -/-, *BRCA1* -/-, *PALB2* mut and 400 cells/ well of wildtype pair was plated; In twenty-four well plate, 500 cells/ well of *BRCA2* -/-, *BRCA1* -/-, *PALB2* mut and 300 cells/ well of wildtype pair was plated. For MEFs, 300 cells per well in six-well plate were seeded. The medium was replaced every 2 days until the colonies were ready for staining in 10–12 days. For staining: Medium was removed from plates, and cells were rinsed with PBS. Fixation was carried out with−Water: Ethanol: Acetic acid (5:4:1) for 30 min followed by staining of colonies with 0.5% crystal violet in Water: Ethanol (3:2) for 2 h at room temperature. The plates were rinsed with water and left for drying overnight at room temperature. Colonies were then counted and IC50 was calculated using results from 3 or more independent experiments. Response curves are shown as mean colony formation +/- S.E.M.

## Western blot analyses

Cells were harvested using−cell scraper and washed in phosphate-buffered saline (PBS), then lysed in 1 x RIPA Lysis and Extraction Buffer (Thermo Scientific #89901) containing Halt™ Protease and Phosphatase Inhibitor Cocktail (Thermo Scientific #1861281). Protein concentration was measured using Pierce™ BCA Protein Assay Kit (Thermo Scientific #23225). The proteins were separated by electrophoresis using 4−20% Mini-PROTEAN® TGX™ Precast Protein Gels (Bio-Rad #4561096) and transferred to Immuno Blot PVDF Membrane (Bio-Rad #1620177). Membranes were blocked using 5%-milk-TBST (or 5%-BSA-TBST for phospho-proteins) for 1 h at room temperature and washed with TBST prior to antibody incubation. Primary antibodies were incubated overnight at 4 °C in 1%-milk-TBST (or 2%-BSA-TBST for phospho-proteins). γ-H2AX (p Ser139) was detected using antibody (NOVUS #NB100-384) diluted 1:2000. Cleaved PARP was detected using antibody (Cell Signaling #9546) diluted 1:2000. Gapdh was detected using Gapdh (14C10) rabbit monoclonal antibody (Cell Signaling Technology #2118) diluted 1:4000. Membranes were washed

## Table 1 | crystallographic table

| Data Collection and processing | |
| --- | --- |
| Space group | P3121 |
| Cell dimensions | |
| a,b,c (Å) | 171.4, 171.4, 63.2 |
| α, β, γ (°) | 90, 90, 120 |
| Resolution (Å) | 42.9-3.24 (3.34–3.24) |
| R$_{merge}$ | 0.264 (2.376) |
| CC1/2 | 0.996 (0.422) |
| I/sigma(I) | 7.9 (1.0) |
| Completeness (%) | 98.4 (86.8) |
| Total observations | 295099 (12119) |
| Unique observations | 16950 (1251) |
| Redundancy | 17.4 (9.7) |
| Refinement Statistics | |
| Resolution (Å) | 42.9 – 3.24 |
| R$_{work}$/R$_{free}$ (%) | 22.80/24.40 |
| No. atoms | 5514 |
| B factors (Å$^2$) | 121 |
| r.m.s deviations | |
| Bond lengths (Å) | 0.002 |
| Bond angles (°) | 0.511 |

with TBST prior to secondary antibody incubation for 1 h at room temperature using respective Goat anti-Rabbit IgG (H + L) HRP (Invitrogen #31466) or Goat anti-Mouse IgG (H + L) HRP (Invitrogen #31430)–tagged Secondary Antibody diluted 1:5000 in 2%-milk-TBST. Membranes were washed using TBST and then treated with Amersham™ ECL™ Prime Western Blotting Detection Reagent (Cytiva #RPN2232) for 5 min and images obtained using the Bio-Rad ChemiDoc Imaging System.

## Immunofluorescence (IF) and confocal microscopy
Cells were plated on six well plates with glass-coverslips and treated with RTX-161 day after plating. Four days after treatment, cells were fixed with 4% (v/v) paraformaldehyde for 20 min at 4 °C, washed with PBS, permeabilized with 0.5% (v/v) Triton X for 10 min and blocked with PBS containing 3% BSA. Cells were incubated with primary antibody overnight at 4 °C followed by 3x washes with PBS and then 1 h incubation with secondary antibody. The incubations were performed in the dark in a humidified chamber. After 3x washing in PBS for 3 min, slides were mounted in 20 ul Prolong antifade with DAPI (Life-Technologies) to counterstain the nuclei. Cells were visualized and imaged using Nikon A1R Confocal microscope at a 63X objective magnification, and images were analyzed using ImageJ software. For quantification, >50 cells were counted for all conditions from three independent experiments. The primary antibody used for IF was rabbit anti-gamma H2AX [p Ser139] antibody (Bethyl Lab #A700-053) 1:500 dilution in 1% BSA in PBS. The secondary antibody was Goat anti-Rabbit IgG (H + L) Secondary Antibody, DyLight 488 (Thermo #35552) 1:2000 dilution in 1% BSA in PBS.

## Apoptosis assay
DLD1 *BRCA2* -/-, DLD1 Parental cells were plated on six well plates and treated with RTX-161 1 day after plating. Medium was replaced every 2 days with RTX-161. Cells were harvested 6 days after plating for apoptosis assay using FITC active caspase-3 apoptosis kit (BD biosciences, catalog # 550480) according to manufacturer's instructions and analyzed by FACS with Celesta (Becton Dickinson).

## Endonuclease protection assay
EcoRI and KpnI endonucleases assay: 25 nM 5'-$^{32}$P radiolabeled primer-template DNA Sc15-EcoRI (5'-$^{32}$P LM3-EcoRI/LM4-EcoRI) or Sc19-KpnI (5'-$^{32}$P LM11-Kpn/LM12-Kpn) in reaction buffer (25 mM Tris-HCl pH 7.5, 10 mM MgCl₂, 10% glycerol, 0.1 mg/mL BSA, 0,01% igepal, 1 mM DTT) were incubated in the presence or in the absence of 50 nM Polθ-pol, 100 µM ddCTP, and 10 µM inhibitors, as indicated, for 5 min at room temperature. Next, 0.08 U/µL EcoRI or KpnI together with 500 nM "trapping" DNA Sc14 (RP559/RP702) were added and incubated for further 0–40 min at 30 °C as indicated. 10 nM PolθΔcen and FL-Polθ and 25 nM Sc15-EcoRI and Sc19-Kpn were probed with 0.04 U/µL EcoRI or KpnI (both from New England Biolabs) using the same conditions. Reactions were stopped by addition of formamide loading buffer with 50 mM EDTA, resolved in denaturing 20% urea PAAG and analyzed using Typhoon PhosphorImager (GE Amersham). Quantification was done in ImageQuant TL software. All quantified experiments were performed in triplicates and plotted as mean with ± s.d using GraphPad Prism 9 software.

## MMEJ in vitro assays
For the MMEJ in vitro assay in the presence of RTx-161, 10 nM Polθ-pol was first mixed with DMSO (3% final concentration) or RTx-161 at indicated final concentrations 4–500 nM in reaction buffer (25 mM Tris-HCl pH 7.5, 10 mM MgCl₂, 10% glycerol, 0.1 mg/mL BSA, 0,01% IGEPAL CA-630, 1 mM DTT) followed by incubation for 5 min at room temperature. MMEJ reactions were started by the addition of 50 µM dNTP and 25 nM 5'-$^{32}$P radiolabeled pssDNA (RP343/$^{32}$P-RP344), incubated for 15 min at 37 °C, stopped by the addition of denaturing formamide loading buffer with 50 mM EDTA, resolved in denaturing 20% PAGE and analyzed using Typhoon PhosphorImager (GE Amersham). Quantification was performed using ImageQuant TL software. For the MMEJ in vitro assay in the presence of MC160385, 0.2–10 µM MC160385, 10 nM Polθ-pol, 60 nM 5'-$^{32}$P radiolabeled pssDNA and 100 µM dNTPs were used. The reaction was performed as above and stopped by the Proteinase K treatment in 50 mM EDTA and 0.5% SDS. The products were resolved in native 12% PAGE at room temp and analyzed as above.

## Single nucleotide addition assays
Relative rate of Polθ-pol dAMP addition. The relative rates of dAMP incorporation by Polq-pol in the presence of DMSO or MC160385 with various concentrations of dATP were measured using the following primer extension assay in vitro. 2 nM or Polθ-pol was incubated with 5' $^{32}$P-γ-ATP labeled 100 nM primer-template (RP469D/RP486) in 1X buffer (25 mM TrisHCl pH 7.8, 0.01% NP40, 10% glycerol, 1 mM DTT, 10 mM MgCl₂, 0.1 mg/ml BSA) at room temp. 2.5% DMSO, 7.5 nM or 12 nM of MC160385 was added. Reactions were than initiated by the addition of 20, 50, 100, 200, or 300 µM dATP at room temp. Aliquots of the reactions were removed and terminated by the addition of 50 mM EDTA at 3, 3.5, 4 and 5 min after the reaction was initiated. Radio-labeled DNA was then resolved via denaturing 20% PAGE and visualized by Typhoon PhosphorImager (GE Amersham). Rates of dAMP incorporation were calculated from % extension of the prime-template which was determined using ImageJ software. Relative rate of ddCMP addition on DNA/DNA and DNA/RNA. 25 nM nucleic acid DNA/DNA or DNA/RNA primer-template (RP559/RP702 or RP559/RP702R) was mixed with 1X reaction buffer (20 mM TrisHCl pH 7.5, 0.01% NP-40, 0.5 mM DTT, 10% glycerol, 0.1 mg/ml BSA, 10 mM MgCl₂) in the presence of 250 µM ddCTP at room temp. Next, 2 µM RTx-161 or 2.5% DMSO was added. Reactions were initiated by the addition of 10 nM Polθ-pol. Reaction aliquots were removed and terminated by the addition of 50 mM EDTA at the indicated time points. Radio-labeled DNA was then resolved via denaturing 20% PAGE and visualized by Typhoon PhosphorImager. Percent primer extension by single ddCMP addition at each time point was calculated from % extension of the prime-template which was determined using ImageJ software.

## Primer extension assays on DNA/DNA versus DNA/RNA
25 nM DNA/DNA (RP559/RP702) or DNA/RNA (RP559/RP702R) 5'-$^{32}$P labeled templates were used to perform primer extension assays in the presence or in the absence of 2 µM RTx-161 in reaction buffer (20 mM Tris-HCl pH 7.5, 10 mM MgCl₂, 10% glycerol, 0.1 mg/ml BSA, 0.01% IGEPAL CA-630, 0.5 mM DTT). 250 µM ddCTP was mixed with 25 nM primer-template in the presence of 2.5% DMSO or 2 µM RTx-161. The reactions were initiated at room temp by the addition of 10 nM of Polθ-pol and aliquots of the reaction were terminated at the indicated time points by the addition of 25 mM EDTA and 45% formamide. DNA was resolved by denaturing 20% urea PAGE and analyzed using Typhoon PhosphorImager (GE Amersham). Quantification of primer extension was performed using ImageQuant TL software.

## IC$_{50}$ biochemical assay
The following Cy5 fluorescence assay was used to measure the ability of compounds to inhibit Polθ-pol in vitro. The fluorescent based assay was performed as follows: 60 nM of the pre-annealed primer-template containing a 5' Cy5 fluorophore conjugated template strand (RP486-Cy5), a downstream complementary oligo conjugated with a 3' Iowa BlackReg Dark quencher (RP343BHQRQ) and a primer strand (RP469D) was mixed with 50 uM 2'-deoxyribonucleoside

triphosphates (dNTPs), 0.1 mg/mL Bovine serum albumin (BSA), 0.01% NP-40, 10% glycerol, 1 mM dithiothreitol (DTT), 10 mM MgCl₂, 25 mM TrisHCl pH 7.8 in the presence of 2.5% DMSO with or without various concentrations (7-point dilution series) of Polθ-pol small-molecule inhibitors at 37 °C in a volume of 40 μL. The reactions were initiated by the addition of 5 nM of purified recombinant human Polθ-pol (comprising amino acid residues 1792–2590). The reactions were terminated by the addition of 20 mM EDTA after 18 min, and the Cy5 fluorescence intensity was measured using a CLARIOstar (BMG Labtech) plate reader. Reactions were performed in triplicate and the % inhibition at each concentration of the respective compound of Formula I was based on the mean. The $IC_{50}$ of each compound represents the average concentration of compound that resulted in 50% inhibition of Polθ-pol enzymatic activity which was determined from a scatter plot (% inhibition versus compound concentration) curve generated by GraphPad Prism 9 software for each compound inhibition data set.

## Nucleic acids

Radioactively or fluorescently labeled DNA/DNA and DNA/RNA primer-templates or pssDNA (partially single stranded DNA) used in in vitro assays were obtained by annealing labeled vs unlabeled oligonucleotides in a ratio of 1:1.5 using 100 °C–25 °C cooling down conditions. All the oligonucleotides were purchased from IDT. $5'-^{32}P$ radiolabeling was done using T4 polynucleotide kinase (New England Biolabs) and $\gamma^{-32}P$ ATP (Perkin Elmer). The sequences of the oligonucleotides (5′-3′) are listed below: LM3-EcoRI CGGCGAATTCGCAG; LM4-EcoRI AGTAAGAGAACGCTGCGAATTCGCCG; LM11-Kpn CCGGGGTACCGCAG; LM12-Kpn AGTAAGAGAACGCTGCGGTACCCCGG; RP559 CGACGTCGCAG; RP702 GCGCGCTGCGACGTCG; RP702R rGrCrGrCrGrCrUrGrCrGrArCrGrUrCrG; RP343 CTAAGCTCACAGTG; RP343BHQRQ CTAAGCTCACAGTG/3IAbRQSp/; RP344 CACTGTGAGCTTAGGGTTAGCCCGGG; RP469D CTGTCCTGCATGATG; RP486 CACTGTGAGCTTAGTCACATTTCATCATGCAGGACAG; RP486-Cy5 /5Cy5/CACTGTGAGCTTAGTCACATTTCATCATGCAGGACAG; RP494D TTTTCTGCGCGCTGCGACGTCG RP494R rUrUrUrUrCrUrGrCrGrCrGrCrUrGrCrGrArCrGrUrCrG.

## Statistical analysis, reproducibility, and software

Data are expressed as mean ± SEM from at least 3 independent experiments with triplicates for each condition unless stated otherwise. When conducting comparison between two groups, two-tailed unpaired $t$-test was used. Significance was assumed at $p < 0.05$. Asterisks in the figures indicate significance, $^*p < 0.05$, $^{**}p < 0.01$, $^{***}p < 0.001$. Statistically significant $p$-values and number of replicates are indicated in the Figure legends. Combenefit® software version 2.021 was used to perform synergy analyses for drug combinations between RTx-161 and PARPi. GraphPad Prism 9 software was used to calculate $IC_{50}$.

## Proteins

Polθ-pol, Fl-Polθ, PolθΔcen, and Polδ were purified as described[15,23]. Recombinant Polβ and Polλ were provided by the late Dr. Wilson (NIEHS). Dr. Copeland (NIEHS) provided recombinant Polγ. Dr. O'Donnell (Rockefeller University) provided recombinant Polε and Polα. Dr. Arora (Fox Chase Cancer Center) provided recombinant Polκ.

## Chemicals

Olaparib, Rucaparib and Talazoparib were purchased from Selleck Chemicals.

## Reporting summary

Further information on research design is available in the Nature Portfolio Reporting Summary linked to this article.

## Data availability

The X-ray crystallography structure of PolθΔL-pol bound to DNA/DNA, RTx-152 and ddGMP was deposited in the Protein Data Bank (PDB) with the identifier 8GD7. Source data are provided with this paper.

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

## Acknowledgements

The research was supported by: National Institutes of Health grants R01GM137124, R01CA244179, R01GM130889, R35GM152198, and Department of Defense grant W81XWH2010031 to R.T.P.; National Institutes of Health grants R01CA214799, R01CA255360, and R01GM135293 to N.J.; National Institutes of Health grants R01CA244179, R01CA186238 and Leukemia and Lymphoma Society grant TRP 6628-21 to T.S. We thank Dr. Xia (Rutgers University) for EUFA1341 *PALB2* mut and *PALB2* wildtype cells. We thank the late Dr. Wilson (NIEHS) for recombinant Polβ and Polλ. We thank Dr. Copeland (NIEHS) for recombinant Polγ. We thank Dr. O'Donnell (Rockefeller University) for recombinant Polε and Polα. We thank Dr. Arora (Fox Chase Cancer Center) for recombinant Polκ. We acknowledge Rasayan Inc. and Sapala Organics Pvt. Ltd. for chemical synthesis.

## Author contributions

The study was conceived and managed by R.T.P. The manuscript was written by R.T.P. and X.S.C. and co-authors contributed to editing and figure development. W.F. performed X-ray crystallography experiments and structure determination and refinement. Cell biology assays were performed by M.T., G.C., M.C., R.B., J.J.K., Y.W., and U.M.V. Biochemical assays were performed by L.M., T.T., T.H., T.R., N.B. and W.A. M.R. and G.M. performed synthetic chemistry. Protein purifications were performed by T.K. and T.H. J.G. performed in vitro ADME assays. T.S. and N.J. provided support for cell biology assays. W.C. provided support for synthetic chemistry.

## Competing interests

R.T.P. is a cofounder and chief scientific officer of Recombination Therapeutics, LLC. X.S.C. and W.C. are co-founders of Recombination Therapeutics, LLC. The other authors declare no competing interests.
