## [Peer Review File · Nature Communications]

Discovery of a Small-Molecule Inhibitor that Traps Pol θ on DNA and Synergizes with PARP InhibitorsREVIEWER COMMENTS

Reviewer #1 (Remarks to the Author):

In this study, the authors have identified and characterized a novel small-molecule inhibitor of DNA polymerase θ (Pol θ), an important DNA damage response protein that provides a promising drug target in BRCA1/2 mutant cancers. The study is comprehensive, combining DNA synthesis assays, cellular data, structural characterization of Pol θ -inhibitor complexes and biochemical assays designed to elucidate the mechanism of inhibition. Here I will focus on the structural and biochemical aspects of the study.

The structure of Pol θ with RTx-152 reveals the inhibitor bound to the fingers domain, which is in a closed conformation. The structure is informative because it shows that the inhibitor occupies the location where the N and O helices would normally reside when the fingers domain is in an open conformation. This observation suggests that the bound inhibitor traps Pol θ in a closed conformation and prevents reversion to an open conformation, necessary for ongoing DNA synthesis. This possibility is confirmed with biochemical experiments, which show that RTx-152 and RTx-161 inhibit dissociation of Pol θ from the DNA substrate for extended periods of time. These experiments are cleverly designed. Notably, the results demonstrate that dissociation is specifically blocked in the post-catalytic, rather than the pre-catalytic, complex, establishing that inhibitors trap the polymerase on DNA while in the closed state.

Overall, the results are convincing and they establish a new mechanism for small-molecule inhibition of DNA polymerase activity. The paper is very well written for a general audience. I support publication in the present form.

Reviewer #2 (Remarks to the Author):

The authors present evidence that a group of Pol Theta allosteric inhibitors in combination with PARP inhibitors lead to a reduction in generation of PARPi resistance in a number of clinically significant cell lines. This supports the concept of using Pol theta inhibitors and PARPi in combination to treat a number of different cancers.

The mechanistic studies using EcoRI and KPN1 footprinting are consistent with the RTx- compounds trapping the nucleic acid duplex on Pol theta as a possible mechanism of inhibition. Inhibition occurs on DA but not RNA/DNA substrates.

The structural analyses of the Pol Theta inhibitor complexes with the RTx- investigational appears to be properly done. The figures illustrate the authors' points but in multiple cases don't take advantage of optimal choices for contrasting the side chain structures when being compared such as the rightmost panel of Figure 3d. Perhaps having different widths of the bonds shown in two structures would be more contrasting, or a more distinct color choice?

The RMSD values calculated from comparison of bound and unbound pocket regions are very high, often $>5 \text{ \AA}$, suggesting that they are calculated using all atoms. The authors should specify which atom types and residues are being used for the comparison and consider calculating using only C α atoms only (more conventional, usually from $\sim 1-3 \text{ \AA}$), which gives a better idea of the degree of main-chain movements. With all atoms, a single side chain or loop displacement can lead to large RMSD values when using all atoms.

In several places confirmation is used in place of conformation.

Reviewer #3 (Remarks to the Author):

In this manuscript the authors describe the development of a new Poltheta inhibitor, which selectively inhibits Pol θ polymerase (Pol θ -pol) in the closed conformation on B-form DNA/DNA and traps it on DNA. In vitro seems more potent than some of the previously described pol theta inhibitors and it selectively kills HR-deficient cells as well as PARPi resistant cells.

The data provided are convincing and solid. Nevertheless, to justify the need of new inhibitor, the authors have to expand their comparison with other potent pol theta inhibitors and in cell survival assays.

Reviewer 1:

...Overall, the results are convincing and they establish a new mechanism for small-molecule inhibition of DNA polymerase activity. The paper is very well written for a general audience. I support publication in the present form.

Authors response: We thank the referee for their review and insight.

Reviewer 2:

...The structural analyses of the Pol Theta inhibitor complexes with the RTx- investigational appears to be properly done. The figures illustrate the authors' points but in multiple cases don't take advantage of optimal choices for contrasting the side chain structures when being compared such as the rightmost panel of Figure 3d. Perhaps having different widths of the bonds shown in two structures would be more contrasting, or a more distinct color choice?

Authors response:

We thank the referee for their professional review and insight. We have now changed the widths of bonds and changed the colors to enhance contrast in the structure figures as suggested. Figures 3a, 3b, 3d, 3e-f, 3h, and 4a have been modified.

Reviewer 2:

The RMSD values calculated from comparison of bound and unbound pocket regions are very high, often >5 Å, suggesting that they are calculated using all atoms. The authors should specify which atom types and residues are being used for the comparison and consider calculating using only C α atoms only (more conventional, usually from ~1-3 Å), which gives a better idea of the degree of main-chain movements. With all atoms, a single side chain or loop displacement can lead to large RMSD values when using all atoms.

Authors response:

The reviewer made a good point in suggesting we use only the C α atoms for the RMSD values. We have redone the RMSD calculations using only the C α atoms, and we now use those values when RMSD is mentioned in the paper instead. We have also made it clearer within the paper which atoms we have used for these calculations.

Reviewer 2:

In several places confirmation is used in place of conformation.

Authors response:

We have now corrected this spelling error in multiple instances throughout the manuscript. We hope the reviewer will agree that the revised manuscript is now more thorough and clear, and acceptable for publication.

Reviewer 3:

In this manuscript the authors describe the development of a new Poltheta inhibitor, which selectively inhibits Pol θ polymerase (Pol θ -pol) in the closed conformation on B-form DNA/DNA and traps it on DNA. In vitro seems more potent than some of the previously described pol theta inhibitors and it selectively kills HR-deficient cells as well as PARPi resistant cells.

The data provided are convincing and solid. Nevertheless, to justify the need of new inhibitor, the authors have to expand their comparison with other potent pol theta inhibitors and in cell survival assays.

Authors response:

We thank the referee for their professional and fair review and agree that the data provided are convincing. We believe there are many reasons for justifying the need of a new Pol-theta inhibitor. First and foremost, although Pol-theta is an attractive precision oncology drug target for BRCA-deficient cancers and overcoming PARPi resistance, no Pol-theta inhibitors have been FDA approved. Secondly, there is a very high demand for more effective treatments for patients with aggressive cancers that are defective in homologous recombination (HR)-this includes subsets of breast, ovarian, prostate and pancreatic cancers. The fact that there are 4 FDA

approved PARP inhibitors (PARPi) and at least another 2-3 next-generation PARPi being developed by large pharmaceutical companies exemplifies the high demand for new effective cancer treatments in this space. Therefore, if only one PARPi was approved for a single or even two or three indications, this would not meet the high demand for treatments. Clinical studies take many years due in large part to the extended periods of time it takes to recruit patients for clinical studies. Therefore, having multiple effective clinical grade drugs in this space will facilitate potential approvals by enabling many different clinical trials. Lastly, each respective compound has specific physiochemical properties which can enhance or reduce drug-like properties. For example, since we have submitted this manuscript, we have developed an analog of RTX-161 (RTX-284) that exhibits 98% oral bioavailability, superior metabolic stability (>50 min $t_{1/2}$ in liver microsomes) and superior half-life (>5 hr) in plasma compared to other Pol-theta inhibitors in the literature (i.e. ART558, ART812, RP6685). The absorption and pharmacology of inhibitors can make a significant difference in their effects in patients. Potency is only one factor that contributes to efficacy. If we never developed RTX-161 because there were other Pol-theta inhibitors already characterized, we would have never been able to develop our new analog which may represent the best-in-class due to superior pharmacokinetics. We hope to publish a follow-up paper on RTX-284 as we further develop and test this new analog.

We agree with the reviewer that a direct comparison of our inhibitor class to others in the literature will be informative regarding relative cellular activities. We therefore have now performed identical cellular IC₅₀ assays for the commercially available Pol-theta inhibitors, ART558 and RP6685, that were recently published and include this data in Supplementary Figure X. The results show that our compound class (i.e. RTX-161) has similar or slightly better potency in BRCA-deficient cells compared to ART558 and RP6685 using identical assay conditions. We thank the reviewer for their insight and hope they will agree that the revised manuscript is now even more comprehensive with new inhibitor comparison assays, and acceptable for publication.

REVIEWERS' COMMENTS

Reviewer #2 (Remarks to the Author):

On line 205, change X-crystallography to X-ray crystallography.

RMSD values should be rounded to one digit after the decimal point, and CA should be Ca. Multiple instances. Example 1.375 Å (CA residues 2330-2425) should be 1.4 Å (Ca residues 2330-2425).

This is a valuable and informative study in a very important system.